# Entropy Generation for Negative Frictional Pressure Drop in Vertical Slug and Churn Flows

**DOI:** 10.3390/e23020156

**Published:** 2021-01-27

**Authors:** Lei Liu, Dongxu Liu, Na Huang

**Affiliations:** State Key Laboratory of Multiphase Flow in Power Engineering, Xi’an Jiaotong University, Xi’an 710049, China; liudongxul@stu.xjtu.edu.cn (D.L.); huangna1120@stu.xjtu.edu.cn (N.H.)

**Keywords:** entropy generation, two-phase flow, slug flow, churn flow, frictional pressure drop, thermodynamic irreversibility

## Abstract

It is widely accepted that the frictional pressure drop is impossible to be negative for pipe flow. However, the negative frictional pressure drops were observed for some cases of two-phase slug and churn flows in pipes, challenging the general sense of thermodynamic irreversibility. In order to solve this puzzling problem, theoretical investigations were performed for the entropy generation in slug and churn flows. It is found that the frictional pressure drop along with a buoyancy-like term contributes to the entropy generation due to mechanical energy loss for steady, incompressible slug and churn flows in vertical and inclined pipes. Experiments were conducted in a vertical pipe with diameter as 0.04 m for slug and churn flows. Most of the experimental data obtained for frictional pressure drop are negative at high gas–liquid ratios from 100 to 10,000. Entropy generation rates were calculated from experimental data. The results show that the buoyancy-like term is positive and responsible for a major part of entropy generation rate while the frictional pressure drop is responsible for a little part of entropy generation rate, because of which the overall entropy generation due to mechanical energy loss is still positive even if the frictional pressure drop is negative in vertical slug and churn flows. It is clear that the negative frictional pressure drops observed in slug and churn flows are not against the thermodynamics irreversibility.

## 1. Introduction

It is widely accepted that the frictional pressure drop is impossible to be negative for pipe flow because the frictional pressure drop usually represents the mechanical energy loss converted to heat. However, the negative frictional pressure drop were observed or discussed by researchers in some cases of two-phase flows such as vertical and near-vertical gas–liquid flows [1,2,3,4], Taylor flow through vertical capillaries [5], vertical two-phase flow with surfactant additive at high gas–liquid ratios [6], and slug flow with high viscosity oil through vertical and deviated pipes [7,8]. These observations challenge the general sense of thermodynamic irreversibility.

For fluids flowing in pipes and channels, the entropy generation results from fluid friction and heat transfer [9,10,11,12]. The mechanical energy loss due to friction effect converts mechanical energy to heat [13]. The total irreversibility is the sum of heat transfer irreversibility and fluid friction irreversibility [14]. Consequently, the entropy generation due to mechanical energy loss is the quantitative expression of fluid friction irreversibility. Generally, the product of frictional pressure drop and volumetric flow rate is the mechanical energy loss for pipe flow. The frictional pressure drop is thus usually considered to be an indication of friction irreversibility for any pipe or channel flow. For example, in the absence of shaft work and heat transfer, the entropy generation due to fluid friction is proportional to frictional pressure drop for pipeline flow [9,15]. From the Clausius inequality which is the mathematic statement of the second law of thermodynamics [16], the entropy generation for any irreversible process is not negative. From these points of views, a negative frictional pressure drop is supposed to be impossible for pipe and channel flows. However, the exceptions still happened in some cases of vertical and near-vertical slug and churn flows.

The normal explanation for the negative frictional pressure drop occurring in vertical slug flow is the falling liquid film in the annulus between the inside of the pipe wall and the rising gas Taylor bubble. Nevertheless, as discussed by Spedding et al. [2], while the downward liquid annulus flow can explain the wide fluctuations observed in the total pressure drop of vertical slug flow, it cannot explain the negative value of the overall average value of frictional pressure drop. The phenomenon of negative frictional pressure drop was studied by Liu [6]. It was experimentally and theoretically shown that the mechanical energy loss in vertical or inclined slug and churn flows is greater than zero in the case of negative frictional pressure drop [6]. Even so, friction irreversibility still awaits for being investigated for the case of negative frictional pressure drop.

Entropy generation is the measure of irreversibility. It is also an attracting issue for two-phase flow. Considerable studies had been carried out by a lot of researchers. Liquid droplets were found to influence the entropy generations in two-phase flow [15,17]. The entropy-based model was proposed for predicting the settling velocity of a falling particle in a particle–fluid mixture [18]. For nanofluid and ferrofluid flows, the entropy generations were studied with two-phase methods [19,20,21,22]. Based on separated two-phase flow model, several approaches were proposed for estimating the entropy generations in adiabatic two-phase flows [23,24]. In some cases, entropy can be used to predict two-phase flow parameters. The void fraction of two-phase flow was obtained based on the principle of minimum entropy production [25]. Shannon entropy, sample entropy and multi-scale entropy were used to identify two-phase flow patterns [26,27]. However, there is no literature available regarding thermodynamic entropy generation for negative frictional pressure drop in two-phase flow up to the present. Although the falling liquid film sounds to be reasonable for the explanation of negative friction in vertical slug flow, the problem keeps still unsolved because the negative frictional pressure drop leads to the entropy decrease which is usually considered to be against the second law of thermodynamics.

Vertical slug and churn flows at high gas–liquid ratios occur in gas wells and gas-lift reactors, in which the negative frictional pressure drops were observed. By so far, the entropy generation is still a puzzling problem for the negative frictional pressure drops. In order to solve this puzzling problem, the work is performed here for the entropy generation in two-phase slug and churn flows, especially for the cases of negative frictional pressure drop.

## 2. Entropy Generation

The entropy analysis below is conducted from equilibrium thermodynamics combined with fluid dynamics. The two-phase slug flow concerned is assumed to be steady, incompressible, and isothermal. The phases are assumed to remain in thermodynamic equilibrium. The flow in an inclined pipe is considered here as a general case in theory. A vertical pipe is a specific case of inclined pipe corresponding to the inclination angle of 90°. The theoretical results obtained below can be applied to both inclined and vertical flow cases.

### 2.1. Entropy Generation for Two-Phase Flow

The thermodynamic entropy for two-phase flow was derived by Wallis [28] in the particular case of a homogeneous two-phase flow and was considered to be valid for separated two-phase flow. Similar approach was adopted by Hanafizadeh et al. [24]. For steady equilibrium gas–liquid two-phase flow in pipes, the total entropy generation rate is defined as the followings [28]
(1)$${\dot{S}}_{p}=\rho_{H}v_{m}A\frac{d\left(xs_{G}+(1-x)s_{L}\right)}{dz}$$
where *A* is the cross-sectional area of pipe, ${\dot{S}}_{p}$ is the total entropy generation rate per unit length caused by irreversible processes and heat transfer in two-phase flow, *s_G_* is the specific entropy of gas, *s_L_* is the specific entropy of liquid, *v_m_* is the mean velocity of two-phase flow, *x* is gas mass fraction or vapor quality, *z* is the location coordinate along pipe, *ρ_H_* is the homogenous density of two-phase flow, and the product $\rho_{H}v_{m}A$ is the mass flow rate of two-phase mixture.

In Equation (1), the homogenous density is calculated by the following equation
(2)$$\rho_{H}=\frac{x}{\rho_{G}}+\frac{1-x}{\rho_{L}}$$
where *ρ_L_* and *ρ_G_* denote the liquid and gas densities, respectively.

The superficial velocity of gas is equal to the ratio of the gas volume flow rate to the cross-sectional area of pipe, and the superficial velocity of liquid is the ratio of the liquid volume flow rate to the cross-sectional area of pipe. In Equation (1), the mean velocity and the quality are relevant with the superficial velocities of gas and liquid. They can be calculated by the following equations, respectively,
(3)$$v_{m}=v_{SL}+v_{SG}$$
(4)$$x=\frac{\rho_{G}v_{SG}}{\rho_{G}v_{SG}+\rho_{L}v_{SL}}$$
where *v_SL_* and *v_SG_* are the superficial velocities of liquid and gas, respectively.

Slug flow is a kind of separated two-phase flow. The relationship between entropy generation and thermodynamic properties was presented for the separated two-phase flow for which the phases remain in equilibrium [28,29]. Such a relationship can be expressed for steady, incompressible two-phase flow as
(5)$$Td\left(xs_{G}+(1-x)s_{L}\right)=d\left(xh_{G}+(1-x)h_{L}\right)-\frac{1}{\rho_{H}}dp$$
where *h_G_* is the specific enthalpy of gas, *h_L_* is the specific enthalpy of liquid, *T* is the absolute temperature of gas–liquid two-phase flow, and *p* denotes the pressure of flow.

For steady, incompressible two-phase flow in the absence of shaft work and phase change, the energy equation presented in the literature [28,29] is simplified as
(6)$$d\left(xh_{G}+(1-x)h_{L}\right)+g\sin\theta dz=dq$$
where *g* represents the gravitational acceleration, *θ* is the inclination angle to the horizonal, *q* is the heat transfer to the two-phase mixture per unit mass.

From Equations (1), (5) and (6), the entropy generation associated with irreversible processes and heat transfer is expressed as
(7)$${\dot{S}}_{p}T=v_{m}A\rho_{H}\frac{dq}{dz}-v_{m}A\left(\rho_{H}g\sin\theta +\frac{dp}{dz}\right)$$

The first term on the right hand side of Equation (7) obviously is associated with heat transfer. The others on the right hand side of Equation (7), which include the effects of gravity and pressure gradient, is relevant with the irreversibility source in addition to heat transfer. Following the approaches presented [9,13,19,30], the total entropy generation rate is expressed as follows:(8)$${\dot{S}}_{p}={\dot{S}}_{q}+{\dot{S}}_{f}$$
where ${\dot{S}}_{q}$ denotes the thermal entropy generation rate per unit length due to heat transfer, and ${\dot{S}}_{f}$ is the frictional entropy generation rate per unit length due to mechanical energy loss. Quantitatively, ${\dot{S}}_{f}$ is associated with the friction effects converting mechanical energy to heat.

From Equations (7) and (8), the term ${\dot{S}}_{f}$ is supposed to be the following:(9)$${\dot{S}}_{f}T=-v_{m}A\left(\rho_{H}g\sin\theta +\frac{dp}{dz}\right)$$

It should be paid attention that the term $\left(\rho_{H}g\sin\theta +\frac{dp}{dz}\right)$ in Equation (9) is not the frictional pressure gradient because the density is not the actual density of two-phase flow. For single-phase flow in channels, the mechanical energy loss can be related to entropy production [31]. In the subsequent part, it would be proved that the term $-v_{m}A\left(\rho_{H}g\sin\theta +\frac{dp}{dz}\right)$ is related to the mechanical energy loss in slug and churn flows.

### 2.2. Mechanical Energy Loss in Slug and Churn Flows

The entropy generation due to mechanical energy loss is the measure of fluid friction irreversibility. The mechanical energy loss is often termed as energy dissipation. It is induced by the fluid friction effects which become comparatively complex for slug and churn flows because of the distribution and relative motion of liquid and gas phases. Slug and churn flows are also termed as intermittent flow because of the alternate appearance of liquid slug and Taylor bubble. A number of slug flow or intermittent flow models were described in the literature [32,33,34,35,36,37]. These models are applicable to the churn flow in which the Taylor bubble and liquid slug are distorted. Slug flow models usually focus on a slug flow unit which consists of a liquid slug zone and a Taylor bubble zone. For a circular pipe with uniform cross section, the sketch of slug flow unit is illustrated in Figure 1. The pipe inclination angle is assumed to be positive for upward flow and negative for downward flow. The length of a slug flow unit is given by
(10)$$L_{U}=L_{s}+L_{b}$$
where, *L_U_* is the length of slug unit, *L_s_* is the length of liquid slug, and *L_b_* is the length of Taylor bubble.

The entropy generation due to mechanical energy loss would be focused below. Assuming the slug flow is steady, incompressible and isothermal, integrating Equation (9) over a slug flow unit gives
(11)$${\dot{S}}_{f}TL_{U}=v_{m}A\left(\Delta p_{U}-\rho_{H}gL_{U}\sin\theta \right)$$
where *L_U_* is the length of a slug flow unit, and $\Delta p_{U}$ is the pressure drop over a slug flow unit.

In practice, the right hand side of Equation (11) is the mechanical energy loss of a slug flow unit. On the basis of slug flow models [33,34,35,36,37], the mechanical energy loss was derived by Liu [6] for inclined and vertical slug flow, given by
(12)$$\begin{array}{l} v_{m}A\left(\Delta p_{U}-\rho_{H}L_{U}g\sin\theta \right)\\ \begin{array}{cc} & = \end{array}\frac{1}{8}f_{Lb}\rho_{L}v_{Lb}^{2}\left|v_{Lb}\right|B_{Lb}L_{b}+\frac{1}{8}f_{Gb}\rho_{G}v_{Gb}^{2}\left|v_{Gb}\right|B_{Gb}L_{b}\\ \begin{array}{cc} & + \end{array}\frac{1}{8}f_{ib}\rho_{G}{\left(v_{Gb}-v_{Lb}\right)}^{2}\left|\left(v_{Gb}-v_{Lb}\right)\right|B_{ib}L_{b}\\ \begin{array}{cc} & + \end{array}\frac{1}{8}f_{s}\rho_{s}v_{m}^{3}\pi DL_{s}+\rho_{L}H_{Ls}\left(\frac{H_{Ls}}{H_{Lb}}-1\right){\left(v_{t}-v_{m}\right)}^{2}v_{m}A \end{array}$$
where *B_Lb_* and *B_Gb_* denote the wetted perimeters of gas and liquid at pipe wall for Taylor bubble, respectively; *B_ib_* is the perimeter of gas–liquid interface at the cross-sectional area of Taylor bubble, *D* denotes the diameter of circular pipe, the symbols *f_Gb_*, *f_Lb_* and *f_ib_* are the Darcy friction factors for gas, liquid, and interface in Taylor bubble, respectively; *H_Lb_* and *v_t_* are the liquid holdup and the translational velocity of Taylor bubble, respectively; *v_Gb_* and *v_Lb_* represent the actual velocities of gas and liquid in Taylor bubble, respectively; *H_Ls_*, *f_s_* and *ρ_s_* denote the liquid holdup, the Darcy friction factor and the density for liquid slug, respectively; and *π* represents circumference ratio.

From Equations (11) and (12), the relationship between entropy generation and friction effects is obtained as
(13)$$\begin{array}{l} {\dot{S}}_{f}=\frac{1}{8T}f_{Lb}\rho_{L}v_{Lb}^{2}\left|v_{Lb}\right|B_{Lb}\frac{L_{b}}{L_{U}}+\frac{1}{8T}f_{Gb}\rho_{G}v_{Gb}^{2}\left|v_{Gb}\right|B_{Gb}\frac{L_{b}}{L_{U}}\\ \begin{array}{cc} & + \end{array}\frac{1}{8T}f_{ib}\rho_{G}{\left(v_{Gb}-v_{Lb}\right)}^{2}\left|\left(v_{Gb}-v_{Lb}\right)\right|B_{ib}\frac{L_{b}}{L_{U}}\\ \begin{array}{cc} & + \end{array}\frac{1}{8T}f_{s}\rho_{s}v_{m}^{3}\pi D\frac{L_{b}}{L_{U}}+\frac{1}{T}\rho_{L}H_{Ls}\left(\frac{H_{Ls}}{H_{Lb}}-1\right){\left(v_{t}-v_{m}\right)}^{2}v_{m}A\frac{1}{L_{U}} \end{array}$$

Equation (13) demonstrates that five friction effects lead to the entropy generation rate ${\dot{S}}_{f}$ in two-phase slug flow. On the right hand side of Equation (13), the first term represents the source of entropy generation corresponding to the liquid friction at the pipe wall in Taylor bubble zone. It is obvious that the first term is always positive whether the velocity $v_{Lb}$ is positive or negative. The second term, which is not negative clearly, is the source of entropy generation corresponding to the gas friction at the pipe wall in Taylor bubble zone. The third term, which is obviously impossible to be negative, indicates the source of entropy generation due to the interfacial friction. The fourth term, which is also impossible to be negative since the mean velocity *v_m_* is not less than zero, indicates the source of entropy generation due to the liquid friction at the pipe wall of liquid slug zone. The final term is the source of entropy generation due to the irreversible mixing or acceleration between the front of liquid slug and the tail of Taylor bubble. Such mixing effect is practically a frictional loss arising from vortices. Physically, *H_Ls_* is greater than *H_Lb_* for slug flow and hence the case $\left(\frac{H_{Ls}}{H_{Lb}}-1\right)>0$ is evident. As a result, the final term is not less than zero. Consequently, the entropy generation rate ${\dot{S}}_{f}$, which is the sum of five terms on the right hand side of Equation (13), is impossible to be less than zero. An inequality is thus obtained as
(14)$${\dot{S}}_{f}\geq 0$$

Equation (14) is consistent with the Clausius inequality which is the mathematical statement of the Second law in thermodynamics.

Since ${\dot{S}}_{f}$ represents the overall entropy generation rate per unit length corresponding to mechanical energy loss, the product ${\dot{S}}_{f}TL_{U}$ on the left hand side of Equation (11) is thus essentially the heat power converted from the mechanical energy by different friction effects in a slug flow unit.

The length of a pipe is often not the integral multiple of slug flow unit. Denoting the total pressure drop by Δ*p_t_* for steady slug flow with the pipe length *L*, Δ*p_t_* is related with Δ*p_U_* as the following:(15)$$\frac{\Delta p_{t}}{L}=\frac{\Delta p_{U}}{L_{U}}$$

For steady, incompressible slug or churn flow in pipes, the frictional pressure drop is the subtraction of the gravitational pressure drop from the total pressure drop, expressed by
(16)$$\Delta p_{f}=\Delta p_{t}-\rho_{TP}gL\sin\theta$$
where Δ*p_f_* denotes the frictional pressure drop over the pipe length *L*, and *ρ_TP_* indicates the actual density of two-phase slug flow, defined as
(17)$$\rho_{TP}=\rho_{L}H_{L}+\rho_{G}(1-H_{L})$$
where *H_L_* is the liquid holdup of slug or churn flow in a pipe.

From Equations (11), (15), and (16), the overall entropy generation rate due to mechanical energy loss is given for steady and incompressible slug and churn flows as the followings.
(18)$${\dot{S}}_{f}=\frac{v_{m}A}{T}\frac{\Delta p_{f}}{L}+\frac{v_{m}A}{T}\left(\rho_{TP}-\rho_{H}\right)g\sin\theta$$

Both Equations (13) and (18) are the expressions of entropy generation rate due to mechanical energy loss or friction effects for slug and churn flows in pipes. They are identical with each other. The entropy generation expressed by Equation (13) is associated with the gravity. By contrast, the entropy generation expressed by Equation (18) is independent of the gravity. Equation (13) indicates that the essence of the entropy generation ${\dot{S}}_{f}$ is the frictional dissipation in slug and churn flows. The irreversible components given in Equation (13) can be only determined by numerical computation, and thus Equation (13) is suitable for theoretical modeling. On the while, Equation (18) is applicable for the prediction of entropy generation from experimental data because the frictional pressure drop $\Delta p_{f}$, the two-phase density $\rho_{TP}$, and other quantities on the right side of Equation (18) can be measured in experiments.

### 2.3. Buoyancy-Like Term on Entropy Generation

For turbulent single-phase flow in a pipe, the calculation for the entropy generation rate due to viscous dissipation was presented by Bejan [9]. The calculation is expressed here as the following:(19)$${\dot{S}}_{f}=\frac{1}{T}\frac{\dot{m}}{\rho }\frac{\Delta p_{f}}{L}$$
where $\dot{m}$ is mass flow rate, and *ρ* is fluid density.

The relationships $\dot{m}=\rho_{H}v_{m}A$ and $\rho =\rho_{H}$ are readily achieved for two-phase flow. Comparing Equation (18) with Equation (19), it is found that there is a new source of entropy in addition to the frictional pressure drop $\Delta p_{f}$ for two-phase flow, which is the term $\frac{v_{m}A}{T}\left(\rho_{TP}-\rho_{H}\right)g\sin\theta$. The entropy generation rate as expressed by Equation (18) is not only dependent on the frictional pressure drop, but also associated with the product $\left(\rho_{TP}-\rho_{H}\right)g\sin\theta$.

The term $\left(\rho_{TP}-\rho_{H}\right)g\sin\theta$ is a buoyancy-like effect on the entropy generation for slug and churn flows. This conclusion can be extended to any separated two-phase flow. It looks like that a pseudo fluid with the density *ρ_H_* passes through a pseudo fluid with the density *ρ_TP_* and thus the buoyancy-like effect $\left(\rho_{TP}-\rho_{H}\right)g\sin\theta$ arises. Essentially, the buoyancy-like term is an indication of drag or pressure loss for two-phase flow in vertical and inclined pipes. It should be emphasized here that the buoyancy-like term $\left(\rho_{TP}-\rho_{H}\right)g\sin\theta$ does not mean a buoyancy-driven two-phase flow which often occurs in gas-lifting process.

For horizontal two-phase flow, the entropy generation *S_f_* is only associated with the frictional pressure drop $\Delta p_{f}$ because the buoyancy-like term $\left(\rho_{TP}-\rho_{H}\right)g\sin\theta$ turns to be zero. For homogenous two-phase flow which corresponds to the case $\rho_{TP}=\rho_{H}=\rho$, the buoyancy-like term disappears, and thus Equation (18) reduces to be Equation (19). For single-phase flow, the condition $\rho_{TP}=\rho_{H}=\rho$ is clear and Equation (18) reduces to be Equation (19). The frictional pressure drop is the only source of frictional irreversibility for steady single-phase, homogenous two-phase flow and horizontal separated two-phase flows. It is readily deduced that the negative frictional pressure drop is undoubtedly impossible for single-phase flow and horizontal two-phase flow since the inequality ${\dot{S}}_{f}>0$ is required for irreversible processes by the Second law of thermodynamics.

For vertical and inclined two-phase flows, the buoyancy-like term is positive. The frictional pressure drop is not the only source of frictional irreversibility. The buoyancy-like term is of great significance for expressing frictional irreversibility and entropy generation. The frictional pressure drop along with the buoyancy-like term gives rise to the frictional irreversibility for steady two-phase flow in vertical and inclined pipes. It is evident from Equations (14) and (18) that the inequality required by the Second law of thermodynamics is satisfied if only the inequality $\frac{\Delta p_{f}}{L}+\left(\rho_{TP}-\rho_{H}\right)g\sin\theta \geq 0$ instead of the inequality $\Delta p_{f}\geq 0$ is proved. In other words, the inequality $\Delta p_{f}\geq 0$ is not necessary while the inequality $\frac{\Delta p_{f}}{L}+\left(\rho_{TP}-\rho_{H}\right)g\sin\theta \geq 0$ becomes necessary for a steady and incompressible two-phase flows from the point of view of irreversibility.

Define the entropy generation rates corresponding to frictional pressure drop and buoyancy-like term as the followings, respectively
(20)$${\dot{S}}_{f1}=\frac{v_{m}A}{T}\frac{\Delta p_{f}}{L}$$
(21)$${\dot{S}}_{f2}=\frac{v_{m}A}{T}\left(\rho_{TP}-\rho_{H}\right)g\sin\theta$$

Thus Equation (18) becomes
(22)$${\dot{S}}_{f}={\dot{S}}_{f1}+{\dot{S}}_{f2}$$
where ${\dot{S}}_{f1}$ is the entropy generation rate per unit length due to frictional pressure drop, and ${\dot{S}}_{f2}$ is the entropy generation rate per unit length due to buoyancy-like term.

Equation (22) shows that the overall entropy generation due to mechanical energy loss includes two parts. One is the part of entropy generation due to frictional pressure drop. The other is the part of entropy generation due to buoyancy-like term. The inequalities ${\dot{S}}_{f}\geq 0$ is always necessary as demonstrated by Equation (14). On the while, the inequalities ${\dot{S}}_{f1}\geq 0$ and ${\dot{S}}_{f2}\geq 0$ are not always necessary.

## 3. Experiment

The entropy generation and negative frictional pressure were studied for air–water slug and churn flows at high gas–liquid ratios in the experimental apparatus which was almost the same with the one introduced by Liu [6]. The experiment was conducted in a vertical plexiglas pipe corresponding to the inclination angle *θ* = 90°. The schematic diagram of the apparatus is shown in Figure 2. This apparatus includes a water supply system, an air supply system, a vertical test section, a data acquisition system (DAQ) and some valves. The liquid supply system consists of a pressure transducers (1), a platinum resistance thermometers (2), a gas–liquid separator (3), a small liquid rotometer (4), a medium liquid rotometer (5), a large liquid rotometer (6), and a liquid bypass (7). The air supply system consists of a pressure transducers (1), a platinum resistance thermometers (2), an air compressor (8), a buffer tank (9), a small orifice gas flow meter (10), a medium orifice gas flow meter (11), and a large orifice gas flow meter (12).

The vertical test section consists of a pressure transducers (1), a platinum resistance thermometers (2), a T-junction mixer (13), a check valve (14), a vertical ruler (15), a vertical Plexiglas pipe (16), and a differential pressure transducer (17). The inside diameter and length of the vertical Plexiglas pipe are 0.04 m and 5.6 m, respectively. Air and water were mixed in the T-junction mixer, then generating slug or churn flow.

By adjusting the control valves on both sides of the T-junction mixer, the specified gas–liquid ratio was achieved in the experiment. When the slug or churn flow became steady in the vertical test section, the data were measured with the differential pressure transducer, pressure transducers, platinum resistance thermometers and orifice flow meters shown in Figure 2. In a test run, the data were continuously acquired by a data acquisition system. The superficial gas velocity was calculated in situ for the vertical test section from the gas mass flow rate measured with orifice meter. A liquid rotometer was chosen from the three liquid rotometers to monitor the liquid flow rate. The total pressure drop was measured by the differential pressure transducer over a length of 4 m of the vertical pipe. As the data acquisition had finished, the valves at both sides of the T-junction mixer were quickly closed. The check valve prevented the liquid from flowing downward and a liquid column was trapped in the vertical test section. The height of the liquid column was measured by the vertical ruler and then the mean liquid holdup in the vertical test section was determined, similar to quick-close-valve approach.

The pressure and temperature are approximately 0.12 MPa and 295 K in this experiment. The density is 1.30 kg/m^3^ and the viscosity is 0.0000182 Pa s for air. The density is 998 kg/m^3^ and the dynamic viscosity is 0.00096 Pa s for water. The superficial gas velocities are in the range from 0.5 m/s to 8 m/s and the superficial liquid velocities are between 0.00005 m/s and 0.08 m/s. The Reynolds numbers corresponding to superficial velocities were between 1400 and 23,000 for gas phase. The Reynolds numbers corresponding to superficial velocities were between 2 and 3300 for liquid phase.

The gas–liquid ratios were specified as 100, 500, 1000, 5000, and 10,000 in this experiments. The gas–liquid ratio is defined as the superficial gas velocity divided by the superficial liquid velocity, given by
(23)$$R_{GL}=\frac{v_{SG}}{v_{SL}}$$
where *R_GL_* is gas–liquid ratio.

The accuracy was 0.2 for both the pressure transducers and the differential pressure transducer. The temperature in the vertical Plexiglas pipe was probed by a platinum resistance with a standard uncertainty of 0.1 K. The superficial velocities of gas and liquid were determined by the flow meters with accuracies as 1.0. The liquid holdup was quantified following the quick-close-valve approach. The standard uncertainty for liquid holdup is 0.007. The frictional pressure drop was obtained from Equations (16) and (17) on the basis of experimental data. The standard uncertainty for the frictional pressure drop is 0.079 kPa. The measurement uncertainties were described in detail by Liu [6].

## 4. Results and Discussion

The experimental data obtained for vertical slug and churn flows are listed in Table 1 in which the gas mass fraction *x* has a relationship with *R_GL_* as
(24)$$x=\frac{\rho_{G}}{\rho_{G}+\frac{1}{R_{GL}}\rho_{L}}$$

In Table 1, the superficial velocity *v_SG_*, the total pressure drop per unit length $\Delta p_{t}/L$, and the liquid holdup *H_L_* were measured directly. The gas–liquid ratio *R_GL_* was calculated by Equation (23) in which the superficial velocity of liquid was measured directly. The actual density of two-phase flow was determined from Equation (17) and the homogenous density *ρ_H_* was computed by the Equations (2) and (24).

The frictional pressure drops were calculated by Equations (16) and (17) and the entropy generation rates due to mechanical energy loss were calculated with Equation (18) from the data in Table 1. The mean velocity *v_m_* in Equation (18) was determined by Equations (3) and (23). The results are plotted against superficial gas velocity in Figure 3 and Figure 4.

As shown in Figure 3 in which the horizontal dashed line corresponds to the case Δ*p* = 0, most of the experimental data obtained for frictional pressure drop are negative at high gas–liquid ratios from 100 to 10,000 and at the superficial gas velocity from 0.5 m/s to 8 m/s. As described by Liu [6], the negative frictional pressure drops are not simply attributable to measurement error. If the superficial gas velocity is not more than 2.8 m/s, the two-phase flow patterns are slug flow, and all the frictional pressure drops measured are negative at gas–liquid ratios from 100 to 10,000. When the superficial gas velocities locate between 3.5 m/s and 6.4 m/s, the two-phase flow patterns are churn flow, and both positive and negative values arise for frictional pressure drop. When the superficial gas velocity reaches up to 8 m/s, the two-phase flow patterns are churn flow and all the magnitudes of frictional pressure drop become positive at different gas–liquid ratios. The minimum value of the frictional pressure drop per unit length is −0.29 kPa/m, emerging in slug flow at the velocity *v_SG_* = 1.0 m/s and the gas–liquid ratio *R_GL_* = 100. Both superficial gas velocity and gas–liquid ratio have significant effects on frictional pressure drop.

As shown obviously in Figure 4, all the values obtained for the entropy generation rate due to mechanical energy loss are positive at different gas–liquid ratios and superficial gas velocities. These experimental results verify the inequality expressed by Equation (14) which is consistent with the Clausius inequality. Corresponding to the same gas liquid ratio and superficial velocity, the entropy generation rate ${\dot{S}}_{f}$ is positive even if the frictional pressure drop $\Delta p_{f}/L$ is negative. It is clear that the frictional pressure drop is negative in some cases of vertical slug and churn flows, which is not against the irreversibility required by the Second law of thermodynamics.

As indicated by Figure 4, the gas–liquid ratio has a little effect on the entropy generation rate ${\dot{S}}_{f}$. Otherwise, the superficial gas velocity has a dominant effect on the entropy generation ${\dot{S}}_{f}$. The analysis of variance (ANOVA) was carried out for the data in Figure 3. The F-values are 4.8 and 169 for gas–liquid ratio and superficial gas velocity, respectively. At the 0.01 level of significance, the critical F-values are approximately 4.2 for gas–liquid ratio and 3.7 for superficial gas velocity, respectively. The F-value 4.8 is slightly greater than the critical F-value 4.2 for gas–liquid ratio, indicating barely that gas–liquid ratio is a significant factor for the entropy generation rate ${\dot{S}}_{f}$. The F-value 169 is far more than the critical F-value 3.7 for superficial gas velocity, indicating distinctly that superficial gas velocity is a very significant factor for the entropy generation rate ${\dot{S}}_{f}$. Superficial gas velocity has much more influence on the entropy generation rate ${\dot{S}}_{f}$ than gas–liquid ratio. A correlation is obtained for entropy generation rate and superficial gas velocity by regression as the followings
(25)$${\dot{S}}_{f}=0.0098v_{SG}^{0.74}$$

The correlation coefficient is 0.99 and the prediction interval with probability of 95% is ±0.009 W/(K⋅m) for the regression above.

The magnitudes for ${\dot{S}}_{f1}$, ${\dot{S}}_{f2}$, and ${\dot{S}}_{f}$ were determined by Equations (20)–(22) from the experimental data in Table 1. The comparisons for ${\dot{S}}_{f1}$, ${\dot{S}}_{f2}$, and ${\dot{S}}_{f}$ were plotted in Figure 5 in which the hollow marks correspond to the negative frictional pressure drops while the solid marks correspond to the positive frictional pressure drops.

In Figure 5a–e, the graphs (a)–(e) correspond to the gas–liquid ratios 100, 500, 1000, 5000, and 10,000, respectively. The distributions from the data points for ${\dot{S}}_{f1}$ and ${\dot{S}}_{f}$ look like thumb shapes at such different gas–liquid ratios. It is clear that the entropy generation rate ${\dot{S}}_{f1}$ is negative in the case of negative frictional pressure drop. On the while, both the entropy generation rates ${\dot{S}}_{f}$ and ${\dot{S}}_{f2}$ are positive in the case of negative frictional pressure drop. At gas–liquid ratio 100 as shown in Figure 5a, the frictional pressure drops are negative in the range of superficial gas velocity from 0.5 m/s to 2 m/s. At gas–liquid ratio 10,000 as shown in Figure 5e, the frictional pressure drops are negative in the range of superficial gas velocity from 0.5 m/s to 6.4 m/s. By comparing Figure 5a–e, it is easily recognized that the range of superficial gas velocity becomes wide for negative frictional pressure drop as the increase of gas–liquid ratio. The range of superficial gas velocity increases with the growth of gas–liquid ratio, in which the entropy generation rate ${\dot{S}}_{f2}$ is negative because of negative frictional pressure drop.

In the case $\Delta p_{f}/L<0$, the entropy generation rate ${\dot{S}}_{f}$ is greater than but very close to the entropy generation rate ${\dot{S}}_{f2}$ for the gas–liquid ratios from 100 to 10,000. In the case $\Delta p_{f}/L>0$, the entropy generation rate ${\dot{S}}_{f}$ is observably less than the entropy generation rate ${\dot{S}}_{f2}$ at gas–liquid ratios from 100 to 500, but slightly less than the entropy generation rate ${\dot{S}}_{f2}$ at gas–liquid ratios from 1000 to 10,000. It is also observed from Figure 5 that the data points of ${\dot{S}}_{f}$ versus *v_SG_* have similar trend for both slug and churn flows in the case $\Delta p_{f}/L<0$ at the same gas–liquid ratio, which is shown by the hollow marks for ${\dot{S}}_{f}$. The turning of the trend occurs corresponding to the transition from negative frictional pressure drop to positive frictional pressure drop, not corresponding to the transition from slug flow to churn flow. Such turnings imply that the index 0.74 in Equation (27) is under estimated for ${\dot{S}}_{f}$ in the case $\Delta p_{f}/L<0$ and is over predicted for ${\dot{S}}_{f}$ in the case $\Delta p_{f}/L>0$. Consequently, the superficial gas velocity exerts further influence on the entropy generation rate ${\dot{S}}_{f}$ in the case of negative frictional pressure drop in comparison with the case of positive frictional pressure drop.

The wetted perimeter of gas at the vertical pipe wall is zero, namely,  $B_{Gb}=0$ and $B_{Lb}=\pi D$. On the right hand of Equation (13), the second term is thus zero. Equation (13) is hence reduced to be
(26)$$\begin{array}{l} {\dot{S}}_{f}=\frac{1}{8T}f_{Lb}\rho_{L}v_{Lb}^{2}\left|v_{Lb}\right|\pi D\frac{L_{b}}{L_{U}}\\ \begin{array}{cc} & + \end{array}\frac{1}{8T}f_{ib}\rho_{G}{\left(v_{Gb}-v_{Lb}\right)}^{2}\left|\left(v_{Gb}-v_{Lb}\right)\right|B_{ib}\frac{L_{b}}{L_{U}}\\ \begin{array}{cc} & + \end{array}\frac{1}{8T}f_{s}\rho_{s}v_{m}^{3}\pi D\frac{L_{s}}{L_{U}}+\frac{1}{T}\rho_{L}H_{Ls}\left(\frac{H_{Ls}}{H_{Lb}}-1\right){\left(v_{t}-v_{m}\right)}^{2}v_{m}A\frac{1}{L_{U}} \end{array}$$

The irreversible mixing or acceleration between the front of liquid slug and the tail of Taylor bubble gives rise to a pressure loss [33,35,36]. This pressure loss is a minor loss which corresponds to the last terms in Equations (12), (13), and (25). In practice, the frictional pressure drop measured includes this minor loss. Taking this minor loss into account, the frictional pressure drop is expressed for vertical slug flow and churn flow as
(27)$$\begin{array}{l} \frac{\Delta p_{f}}{L}=\frac{1}{8}f_{Lb}\rho_{L}v_{Lb}\left|v_{Lb}\right|\frac{4}{D}\frac{L_{b}}{L_{U}}+\frac{1}{8}f_{s}\rho_{s}v_{m}^{2}\frac{4}{D}\frac{L_{s}}{L_{U}}\\ \begin{array}{ccc} & & + \end{array}\rho_{L}H_{Ls}\left(\frac{H_{Ls}}{H_{Lb}}-1\right){\left(v_{t}-v_{m}\right)}^{2}\frac{1}{L_{U}} \end{array}$$

The liquid between Taylor bubble and pipe wall moves downwards as a falling film and thus the velocity *v_Lb_* is negative for vertical slug flow. The first term on the right hand side of Equation (26), which refers to the shear stress in the Taylor bubble zone, becomes negative under the condition $v_{Lb}<0$. Accordingly, the case Δ*p_f_* < 0 is possible due to the condition $v_{Lb}<0$. This is the explanation for the negative frictional pressure drops. Such explanation has been described by others [2,6]. Even so, the entropy generation component caused by the friction effect corresponding to the velocity $v_{Lb}$, which is the first term in Equation (25), is still positive. As shown in Figure 4, all the entropy generation rates due to mechanical energy loss are positive. This is consistent with Equation (25) in which ${\dot{S}}_{f}$ is certainly positive because all the components of entropy generation rate due to different friction effects are positive for vertical slug and churn flows.

The percentage contributions of frictional pressure drop and buoyancy-like term to the entropy generation rate due to mechanical energy loss are shown in Figure 6 and Figure 7, respectively. The transition from negative frictional pressure drop to positive frictional pressure drop leads to the falcate distributions of experimental data in both Figure 6 and Figure 7. The percent of ${\dot{S}}_{f1}$ in ${\dot{S}}_{f}$ is between −13% and 20%, which can be seen in Figure 6. The percent of ${\dot{S}}_{f2}$ in ${\dot{S}}_{f}$ is between 80% and 113%, which can be seen in Figure 7. By contrast, the buoyancy-like term is responsible for a major part of the overall entropy generation rate due to mechanical energy loss. On the while, the frictional pressure drop is responsible for a little part of the overall entropy generation rate due to mechanical energy loss. In such situations, the overall entropy generation due to mechanical energy loss is still positive even if the frictional pressure drop is negative in vertical slug and churn flows.

Under the circumstance sinθ=1 for vertical upward flow, from Equations (20)–(22), (25) and (26), the following relationship is deduced,
(28)$$\begin{array}{l} {\dot{S}}_{f2}=\frac{1}{8T}f_{Lb}\rho_{L}\left(v_{Lb}-v_{m}\right)v_{Lb}\left|v_{Lb}\right|\pi D\frac{L_{b}}{L_{U}}\\ +\frac{1}{8T}f_{ib}\rho_{G}{\left(v_{Gb}-v_{Lb}\right)}^{2}\left|\left(v_{Gb}-v_{Lb}\right)\right|B_{ib}\frac{L_{b}}{L_{U}} \end{array}$$

Equation (27) indicates that the entropy generation rate ${\dot{S}}_{f2}$ is only associated with the friction effects at the pipe wall and the gas–liquid interface in Taylor bubble zone. The frictional loss in liquid slug zone and the minor loss between the front of liquid slug and the tail of Taylor bubble have no direct influence on the entropy generation rate ${\dot{S}}_{f2}$. It is amazing that the entropy generation rate contributed by the buoyancy-like term is only dependent upon the friction effects in Taylor bubble zone from vertical slug and churn flows. The inequality $v_{Lb}<0$ leads to the inequality $\left(v_{Lb}-v_{m}\right)<0$ because the mixture velocity *v_m_* is greater than zero. Consequently, the first term on the right hand side of Equation (27) has a positive value. The second term on the right hand side of Equation (27) is obviously not less than zero. As a result, the inequality ${\dot{S}}_{f2}>0$ is certain for vertical upward slug and churn flows in the case $v_{Lb}<0$. Combining this result with Equation (21), it is clear that the actual two-phase density *ρ_TP_* is inevitably greater than the homogenous density *ρ_H_* for vertical upward slug and churn flows from a point of view of entropy generation, which is shown in Table 1. An accurate simulation of the flow performance in Taylor bubble region is therefore of great importance for slug and churn flows.

## 5. Conclusions

From steady, incompressible two-phase slug flow model in vertical and inclined pipes, five terms are responsible for the entropy generation due to mechanical energy loss. Frictional pressure drop along with a buoyancy-like term contributes to the entropy generation due to mechanical energy loss for slug and churn flows. The buoyancy-like term is an indication of drag or pressure loss for two-phase flow in vertical and inclined pipes. The inequality $\frac{\Delta p_{f}}{L}+\left(\rho_{TP}-\rho_{H}\right)g\sin\theta \geq 0$ instead of the inequality $\Delta p_{f}\geq 0$ becomes necessary for slug and churn flows from the point of view of thermodynamic irreversibility.

All the experimental data obtained for the frictional pressure drop in vertical slug flow are negative at high gas–liquid ratios from 100 to 10,000. Part of the experimental data obtained for the frictional pressure drop in vertical churn flow are negative at high gas–liquid ratios from 100 to 10,000. The entropy generation rates due to mechanical energy loss are still positive even if the corresponding frictional pressure drops are negative in vertical slug and churn flows. Superficial gas velocity is a very significant factor for the entropy generation rate due to mechanical energy loss. On the while, gas–liquid ratio has a little effect on the entropy generation rate due to mechanical energy loss. Buoyancy-like term is positive and responsible for a major part of entropy generation rate while frictional pressure drop is responsible for a little part of entropy generation rate, because of which the overall entropy generation due to mechanical energy loss is still positive even if the frictional pressure drop is negative in vertical slug and churn flows. The theoretical and experimental results show clearly that the negative frictional pressure drop occurring in some cases of vertical slug and churn flows is not against the irreversibility required by the Second law of thermodynamics.

The entropy generation due to buoyancy-like term is only dependent upon the friction effects in Taylor bubble zone for vertical slug flow. In the case the liquid flowing downward in the Taylor bubble zone of vertical upward slug flow, the actual two-phase density *ρ_TP_* is inevitably greater than the homogenous density *ρ_H_* from a point of view of entropy generation.

## Figures and Tables

**Figure 1 entropy-23-00156-f001:**
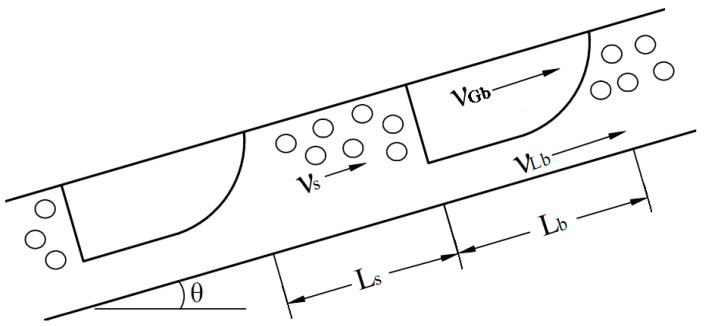
Slug flow unit in an inclined pipe.

**Figure 2 entropy-23-00156-f002:**
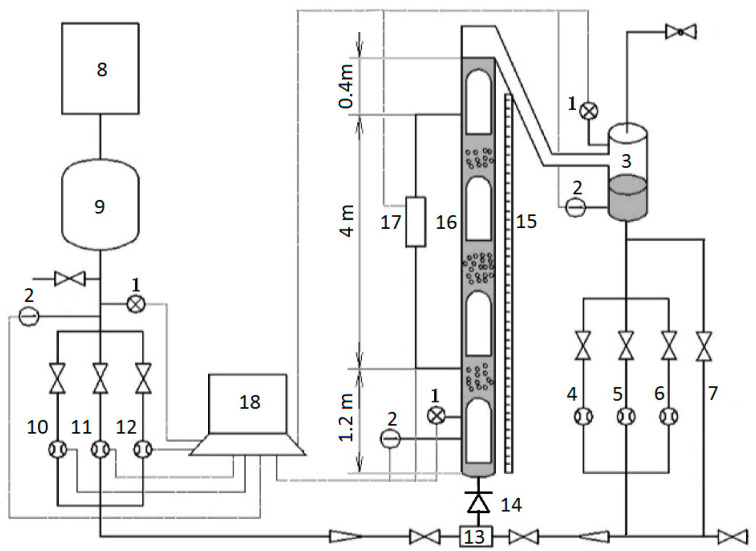
Experimental apparatus. 1. Pressure transducer. 2. Thermometers. 3. Gas–liquid separator. 4. Small liquid rotometer. 5. Medium liquid rotometer. 6. Large liquid rotometer. 7. Liquid bypass. 8. Air compressor. 9. Buffer tank. 10. Small orifice gas flow meter. 11. Medium orifice gas flow meter. 12. Large orifice gas flow meter. 13. T-junction mixer. 14. Check valve. 15. Vertical ruler. 16. Vertical Plexiglas pipe. 17. Differential pressure transducer. 18. Data acquisition system.

**Figure 3 entropy-23-00156-f003:**
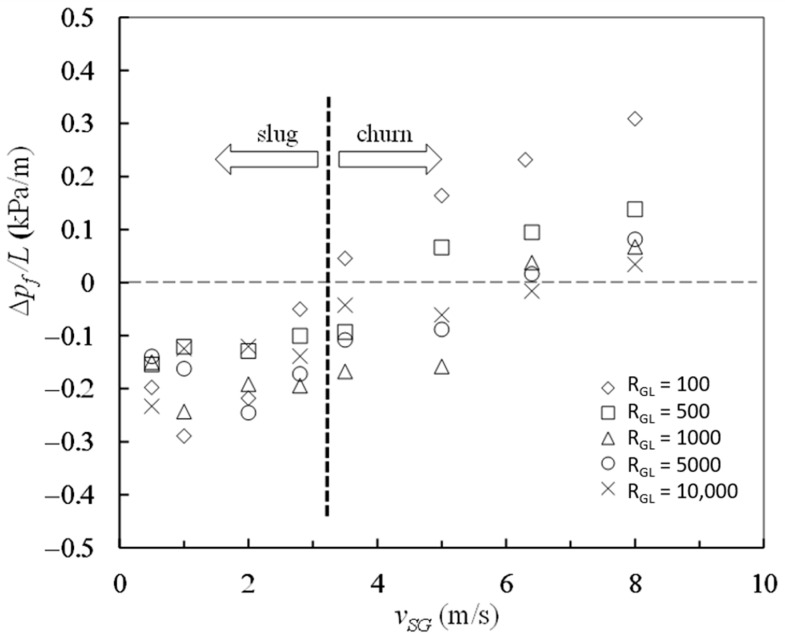
Frictional pressure drops at different gas–liquid ratios and superficial gas velocities.

**Figure 4 entropy-23-00156-f004:**
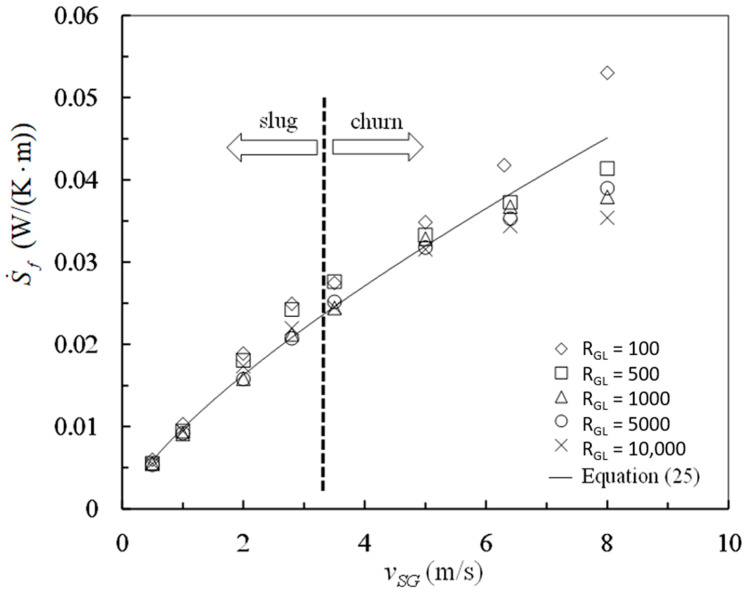
Entropy generation rates due to mechanical energy losses at different gas–liquid ratios and superficial gas velocities.

**Figure 5 entropy-23-00156-f005:**
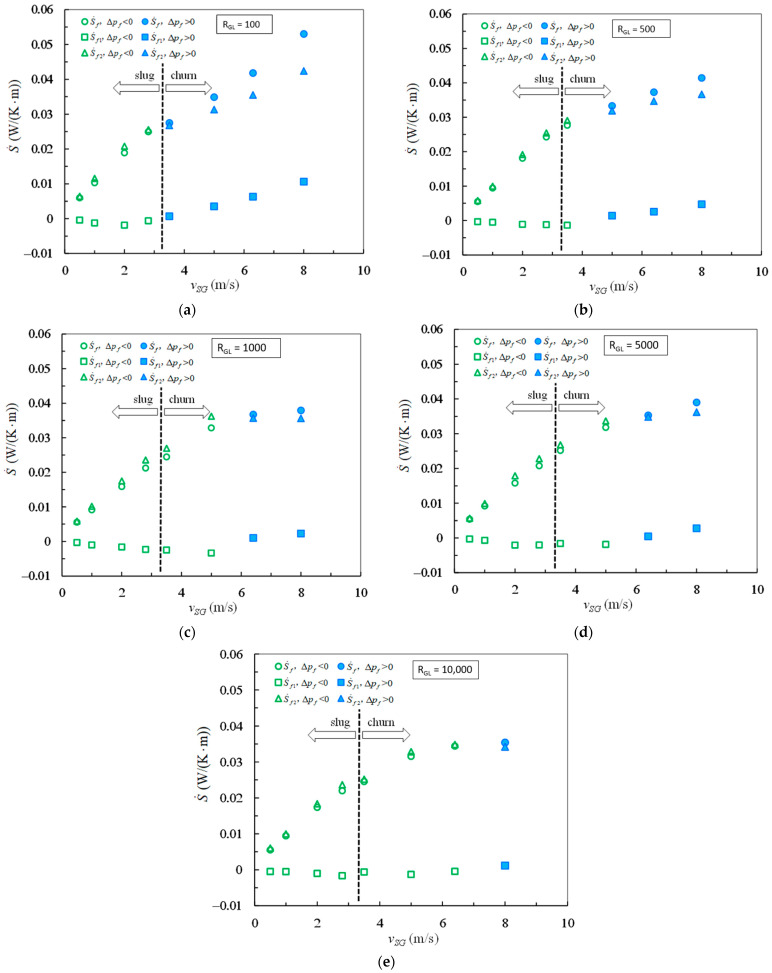
Comparisons for the entropy generation rates ${\dot{S}}_{f1}$, ${\dot{S}}_{f2}$, and ${\dot{S}}_{f}$ (Hollow marks for negative frictional pressure drops and solid marks for positive frictional pressure drops). (**a**) *R_GL_* = 100; (**b**) *R_GL_* = 500; (**c**) *R_GL_* = 1000; (**d**) *R_GL_* = 5000; (**e**) *R_GL_* = 10,000.

**Figure 6 entropy-23-00156-f006:**
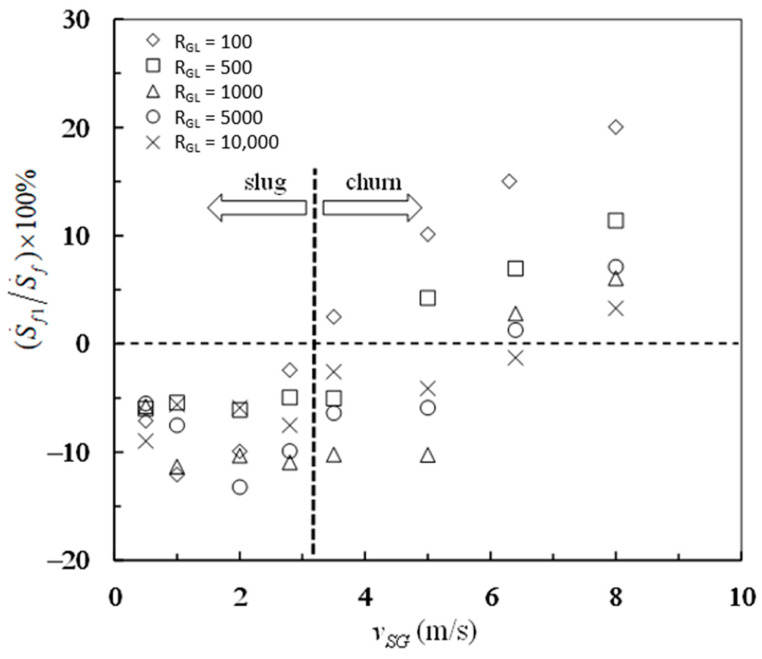
Percentage of the entropy generation rate due to frictional pressure drop in the overall entropy generation rate due to mechanical energy loss.

**Figure 7 entropy-23-00156-f007:**
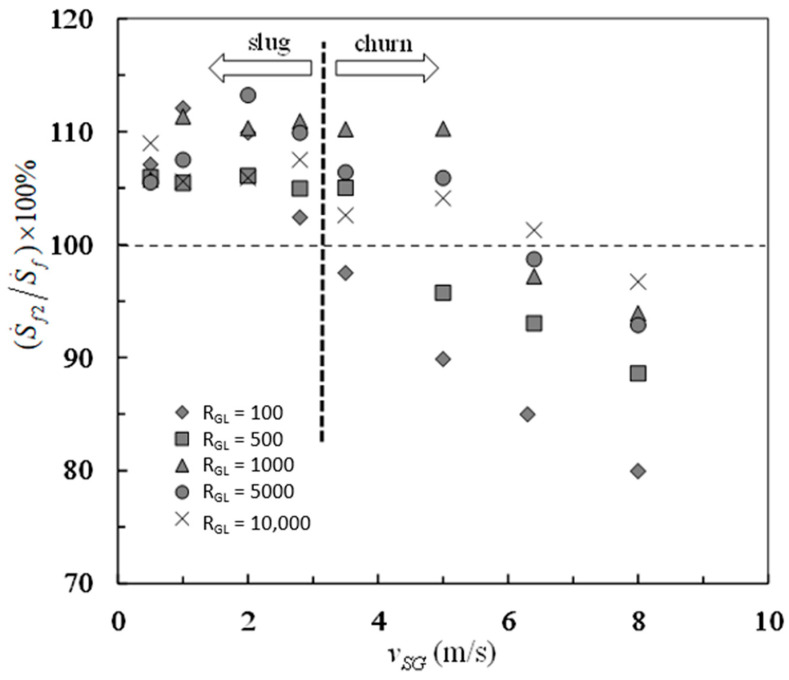
Percentage of the entropy generation rate due to buoyancy-like term in the overall entropy generation rate due to mechanical energy loss.

**Table 1 entropy-23-00156-t001:** Experimental data for vertical air-water slug and churn flows at the temperature 295 K.

*R_GL_*	*v_SG_* (m/s)	Δ*p/L* (kPa/m)	*H_L_*	*ρ_TP_* (kg/m)	*ρ_H_* (kg/m^3^)	Pattern
100	0.5	2.892	0.315	315.3	11.17	slug
100	1.0	2.507	0.285	285.4	11.17	slug
100	2.0	2.305	0.257	257.5	11.17	slug
100	2.8	2.181	0.227	227.7	11.17	slug
100	3.5	1.934	0.192	192.7	11.17	churn
100	5.0	1.730	0.159	159.8	11.17	churn
100	6.4	1.651	0.144	144.8	11.17	churn
100	8.0	1.650	0.136	136.9	11.17	churn
500	0.5	2.613	0.282	282.4	3.29	slug
500	1.0	2.246	0.241	241.5	3.29	slug
500	2.0	2.150	0.232	232.5	3.29	slug
500	2.8	2.061	0.220	220.6	3.29	slug
500	3.5	1.883	0.201	207.6	3.29	churn
500	5.0	1.593	0.155	155.8	3.29	churn
500	6.4	1.397	0.132	132.9	3.29	churn
500	8.0	1.245	0.112	112.9	3.29	churn
1000	0.5	2.627	0.283	283.4	2.30	slug
1000	1.0	2.163	0.245	245.5	2.30	slug
1000	2.0	1.882	0.211	211.6	2.30	slug
1000	2.8	1.801	0.203	203.6	2.30	slug
1000	3.5	1.662	0.186	186.7	2.30	churn
1000	5.0	1.564	0.175	175.7	2.30	churn
1000	6.4	1.369	0.135	136.9	2.30	churn
1000	8.0	1.135	0.108	108.9	2.30	churn
5000	0.5	2.540	0.273	273.4	1.50	slug
5000	1.0	2.175	0.238	238.5	1.50	slug
5000	2.0	1.868	0.215	215.6	1.50	slug
5000	2.8	1.755	0.196	196.7	1.50	slug
5000	3.5	1.702	0.184	178.7	1.50	churn
5000	5.0	1.507	0.162	162.8	1.50	churn
5000	6.4	1.379	0.131	143.8	1.50	churn
5000	8.0	1.159	0.109	109.9	1.50	churn
10,000	0.5	2.612	0.290	290.3	1.40	slug
10,000	1.0	2.233	0.240	240.5	1.40	slug
10,000	2.0	2.051	0.221	221.6	1.40	slug
10,000	2.8	1.857	0.203	203.6	1.40	slug
10,000	3.5	1.660	0.173	173.7	1.40	churn
10,000	5.0	1.495	0.158	158.8	1.40	churn
10,000	6.4	1.276	0.131	131.9	1.40	churn
10,000	8.0	1.053	0.103	104.0	1.40	churn

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
