# Peer review of "Entropy Generation for Negative Frictional Pressure Drop in Vertical Slug and Churn Flows"

_entropy, 2021, doi:10.3390/e23020156_

Round 1

Reviewer 1 Report

The negative frictional pressure drop has been noticed in the results of many two-phase flow studies with few interpretations. In this paper, entropy generation for inclined and vertical slug flow is derived and analyzed. Experiments are carried out and show that negative frictional pressure drop appears at gas-liquid ratios from 100 to 10000. The research is useful.

However, it still lacks interpretations about the switching mechanism. Does the negative frictional pressure drop have anything to do with viscosity or Reynolds number or wall shear stress? Hope more work on it in the future.

Author Response

The suggestions and opinions given by you are valuable for improving the manuscript. We are very grateful for you. We have modified the manuscript

The suggestions and opinions given by you are valuable for improving the manuscript. We are very grateful for you. We have modified the manuscript considering your suggestions and opinions. The changes are highlighted in red in the revised manuscript. The detailed revisions are listed below point by point.

Reviewer 2 Report

Dear author,

Please consider the following remarks.

General remarks

The research object, the goal and objectives of the study are not clearly defined in the INTRODUCTION section.

The author did not explain why the flow in an inclined tube is considered in the theoretical part of the article, and the experiment was carried out in a vertical tube.

The theoretical analysis does not consider the effect of surface tension force on the overall energy balance. When a slug regime is formed from a flow with finely dispersed bubbles, the gas phase's surface area decreases thousands of times, and these forces are released.

The vibration energy of the size of small bubbles has not been estimated. The balance of forces of external and internal pressure of an air bubble does not mean that its size remains unchanged. Due to the inertia of water and the compressibility of air, the bubbles' size fluctuates, sometimes decreasing, then increasing. The energy of these vibrations is an analogue of the internal energy of the flow. When the projectile is formed, this energy is released.

Remarks on References

References are not formatted correctly.

DOI codes are missing, making it difficult for the reviewer to work with the article and make the article less valuable to the reader.

It is recommended to use free Mendeley software with MDPI citation style, available for download in Mendeley Desctop from http://www.zotero.org/styles/multidisciplinary-digital-publishing-institute. To do this, use the menu items in Mendeley Desktop: View - Citation Style - More Styles - Get More Styles - Download.

Remarks on Experiment

It is not clear what the text "were studied with the experimental apparatus introduced by Liu [6]" means. Does this mean that the experiment was carried out on the same apparatus? Or this mean that the experiment was carried out on a similar apparatus? If the experiment was carried out on a different apparatus, then it is necessary to list the differences and explain the reason for choosing another apparatus.

Moreover, it is necessary to provide detailed figures of the experimental setup directly in the article. Detailed drawings of the pressure transducers' location should be provided as the way they are installed can significantly affect the readings of the transducers, especially with multiphase flow.

It is not clear how the flow was formed at the measuring section entrance, whether a honeycomb was used there. Honeycomb is commonly used to align flow in a directional flow and to break large vortices.

It is not clear how accurately the pressure drop was measured. For a correct measurement of the pressure drop, one would have to:

- install a mesh partition with low resistance at the outlet of the measuring section of the pipe, ensuring the transition from the slug flow regime to the normal one (see doi: 10.3390/app10238739);

- place the pressure sensor in the upper part of the pipe after the mesh partition;

- when calculating the pressure drop, subtract the very small but not zero pressure drop across the grate.

Suppose the honeycomb at the inlet and the mesh partition at the outlet of the pipe were absent. In that case, such an experiment's results should be considered important, but preliminary and this should be discussed in the article.

The Reynolds number for the liquid and gaseous phases of the investigated flow should be indicated.

It is not clear how the air bubbles were fed into the water, the supply pressure, and the initial bubble size.

When analyzing theoretical and experimental results, it should be clearly stated that there are different pressure and piezometric lines for gas and liquid phases.

Author Response

The suggestions and opinions given by you are valuable for improving the manuscript. We are very grateful for you. We have modified the manuscript considering your suggestions and opinions. The changes are highlighted in red in the revised manuscript.

Round 2

Reviewer 2 Report

Dear authors, thanks for your corrections and comments. According to the new version of the article, I have no questions or comments.